

# Heavier- and lighter-load isolated lumbar extension resistance training produce similar strength increases, but different perceptual responses, in healthy males and females

James P. Fisher[1], Charlotte Stuart[1], James Steele[1], Paulo Gentil[2] and Jürgen Giessing[3]

[1] School of Sport, Health and Social Sciences, Solent University, Southampton, United Kingdom
[2] Faculty of Physical Education and Dance, Universidade Federal de Goiás, Goiania, Brazil
[3] Institute of Sport Science, University of Koblenz-Landau, Landau, Germany

Corresponding author
James P. Fisher,
james.fisher@solent.ac.uk

## ABSTRACT

**Objectives.** Muscles dominant in type I muscle fibres, such as the lumbar extensors, are often trained using lighter loads and higher repetition ranges. However, literature suggests that similar strength adaptations can be attained by the use of both heavier- (HL) and lighter-load (LL) resistance training across a number of appendicular muscle groups. Furthermore, LL resistance exercise to momentary failure might result in greater discomfort.

**Design.** The aims of the present study were to compare strength adaptations, as well as perceptual responses of effort (RPE-E) and discomfort (RPE-D), to isolated lumbar extension (ILEX) exercise using HL (80% of maximum voluntary contraction; MVC) and LL (50% MVC) in healthy males and females.

**Methods.** Twenty-six participants ($n = 14$ males, $n = 12$ females) were divided in to sex counter-balanced HL ($23 \pm 5$ years; $172.3 \pm 9.8$ cm; $71.0 \pm 13.1$ kg) and LL ($22 \pm 2$ years; $175.3 \pm 6.3$ cm; $72.8 \pm 9.5$ kg) resistance training groups. All participants performed a single set of dynamic ILEX exercise 1 day/week for 6 weeks using either 80% (HL) or 50% (LL) of their MVC to momentary failure.

**Results.** Analyses revealed significant pre- to post-intervention increases in isometric strength for both HL and LL, with no significant between-group differences ($p > 0.05$). Changes in strength index (area under torque curves) were 2,891 Nm degrees 95% CIs [1,612–4,169] and 2,865 Nm degrees 95% CIs [1,587–4,144] for HL and LL respectively. Changes in MVC were 51.7 Nm 95% CIs [24.4–79.1] and 46.0 Nm 95% CIs [18.6–73.3] for HL and LL respectively. Mean repetitions per set, total training time and discomfort were all significantly higher for LL compared to HL ($26 \pm 8$ vs. $8 \pm 3$ repetitions, $158.5 \pm 47$ vs. $50.5 \pm 15$ s, and $7.8 \pm 1.8$ vs. $4.8 \pm 2.5$, respectively; all $p < 0.005$).

**Conclusions.** The present study supports that that low-volume, low-frequency ILEX resistance exercise can produce similar strength increases in the lumbar extensors using either HL or LL. As such personal trainers, trainees and strength coaches can consider other factors which might impact acute performance (e.g. effort and discomfort during the exercise). This data might prove beneficial in helping asymptomatic persons reduce

the risk of low-back pain, and further research, might consider the use of HL exercise for chronic low-back pain symptomatic persons.

## INTRODUCTION

Resistance training (RT) to momentary failure (MF), defined as *"the set end-point when trainees reach the point where despite attempting to do so they cannot complete the concentric portion of their current repetition"* (*Steele et al., 2017b*), appears to produce similar strength adaptations when using both heavier- (HL; >60% 1-repetition maximum; 1RM) and lighter-loads (LL; <60% 1RM) particularly when tested by impartial means (e.g., isometric testing following dynamic training) (*Fisher, Steele & Smith, 2017*; *Fisher, Ironside & Steele, 2017*; *Schoenfeld et al., 2017*). This is argued to potentially result from maximal synchronous, or sequential, motor unit recruitment for HL and LL respectively (*Fisher, Steele & Smith, 2017*), as per the size principle (*Denny-Brown & Pennybacker, 1938*; *Potvin & Fuglevand, 2017*). Furthermore, it has been proposed that complete motor unit recruitment may be a driving factor towards optimizing strength and hypertrophic adaptations (*Fisher, Steele & Smith, 2017*; *Schoenfeld et al., 2014*).

A growing area of research considers the disparity between effort and its perception; *"the amount of mental or physical energy being given to a task"* which is determined by the current ability to meet task demands relative to those demands; and discomfort *"the physiological and unpleasant sensations associated with exercise"* (*Abbiss et al., 2015*; *Marcora, 2009*). A practical example of this is that *"A short maximal voluntary contraction for leg extension, for example, will by nature induce a maximal sense of effort while, initially, other unpleasant sensations will probably be modest. Repeating this maximal contraction several times, however, will increase these unpleasant sensations continuously, whereas the sense of effort will be always the same (i.e., maximal)"* (*Smirnaul, 2012*).

*Fisher, Steele & Smith (2017)* recently hypothesised that disparity in adaptations following RT at HL or LL might be a result of persons anchoring their perceptions of *effort* upon their perceptions of *discomfort*. For example, unless specifically instructed to differentiate the two, most trainees evidently conflate them instead anchoring their perceptions of effort upon their perceptions of discomfort (*Steele et al., 2017d*). Indeed, it has been argued that this can result in trainees underestimating their proximity to MF (*Giessing et al., 2016b*; *Giessing et al., 2016a*) and this effect may be exacerbated under low load conditions (*Steele et al., 2017c*; *Lemos et al., 2017*). A number of studies where participants have been instructed to differentiate their perceptions of effort and discomfort[1] have reported greater discomfort for more fatiguing conditions using lower loads, higher time under load and/or repetitions performed (*Fisher, Ironside & Steele, 2017*; *Fisher, Farrow & Steele, 2017*; *Stuart et al., 2018*). It seems likely that this results from afferent feedback (*Marcora, 2009*) due to a decrease in pH, elevated blood lactate

[1] Using a 0–10 scale for effort and discomfort these have been anchored as 0 = no exertion and 10 = maximal effort, and 0 = no discomfort and 10 = maximal discomfort, respectively.

(BLa), cortisol, and inorganic phosphate ($P_i$) along with increases in $H^+$ as a result of the prolonged elevated adenosine triphosphate (ATP) production (*Genner & Weston, 2014*; *Schott, McCully & Rutherford, 1995*; *Takada et al., 2012*; *MacDougall et al., 1999*).

Perceptions of discomfort appear to be linked to afferent feedback and thus a reason why increased metabolic stress may promote greater discomfort during LL resistance exercise (*Marcora, 2009*). Contrastingly, perceptions of effort are likely related to central motor output to drive motor unit recruitment (*De Morree, Klein & Marcora, 2012*) which may explain similar perceptions of maximal effort when both LL and HL are performed to MF. Considering the potential role of motor unit recruitment in determining adaptation to RT it is important to understand the interactions between load, effort, and discomfort. Within the muscles of the trunk and spine there is a predominance of type I muscle fibres (*Thorstensson & Carlson, 1987*) to sustain the repeated lower force actions needed for postural control and stability. As such, a historical approach has been to train these muscles using LL for a greater number of repetitions.

An area which has received little consideration with respect to the comparison of HL and LL is that of isolated lumbar extension (ILEX) exercise. Low-volume (e.g., single-set) and low-frequency (1 day/week) ILEX exercise is shown to produce considerable strength increases both in trained persons (*Fisher, Bruce-Low & Smith, 2013*; *Steele et al., 2015*) and those symptomatic of chronic low-back pain (*Bruce-Low et al., 2012*; *Steele et al., 2013*; *Steele et al., 2017a*). However, previous studies (*Helmhout et al., 2004*; *Harts et al., 2008*) looking to compare different loading schemes using ILEX RT have used similarly low loads in both HL (35% and 50% of MVC) and LL conditions (20% of MVC). Furthermore, participants did not train to MF, which has been suggested as necessary to standardize RT interventions (*Dankel et al., 2016*). Indeed, it has been noted that there is a lack of research considering different loading patterns for ILEX RT (*Steele, Bruce-Low & Smith, 2015a*), and since the lumbar extensors have been identified as a muscle group that appears to require isolated (e.g., single-joint) exercise to sufficiently strengthen them (*Gentil, Fisher & Steele, 2017*), it seems important to compare HL (>60% MVC) and LL (<60% MVC) RT to MF for ILEX.

Practical treatment approaches using ILEX RT is effective in treating chronic low-back pain (*Steele, Bruce-Low & Smith, 2015a*). As it seems deconditioning is linked to the development of LBP (*Steele, Bruce-Low & Smith, 2014*), strengthening of this musculature using ILEX RT might be an effective preventative approach (*Steele, Bruce-Low & Smith, 2015b*). Furthermore, since strength increases may be related to clinical improvements from ILEX interventions (*Steele et al., 2018*) it seems important to understand the manipulation of RT variables that might optimise strength outcomes. However, to date these loading schemes have not been compared directly, and furthermore, the influence of perceptual responses such as effort and discomfort upon strength outcomes from ILEX exercise has not been considered. It has recently been shown that ILEX RT using LL produces greater acute fatigue and discomfort compared with HL (*Stuart et al., 2018*). As such, when prescribing resistance training for general or clinical populations this might potentiate issues with higher effort exercise (e.g., reaching MF) and result in suboptimal adaptations

**Table 1  Participant characteristics.**

| Variable | HL ($n = 13$) | LL ($n = 13$) |
|---|---|---|
| Age (years) | 23.1 ± 5 | 22.1 ± 2 |
| Stature (cm) | 172.3 ± 9.8 | 175.3 ± 6.3 |
| Body mass (kg) | 71.0 ± 13.1 | 72.8 ± 9.5 |

**Notes.**

HL,  Heavier load (80% MVC);  LL,  lighter load (50% MVC);  MVC,  maximal voluntary contraction.

over a training intervention. Thus, it seems important to examine the effects of loading schemes during ILEX exercise upon these variables.

With the above in mind, the aim of this study was to compare the strength increases of healthy males and females performing ILEX exercise using either HL (80% MVC) or LL (50% MVC) as well as the perceptual responses of effort and discomfort.

## METHODS

A randomised trial was adopted with two experimental groups examining the effects of load during ILEX training in both males and female participants. The primary outcome was ILEX strength changes, and a secondary outcome was perceptual responses. Approval was granted from the University Health, Exercise, and Sport Science (HESS) ethics committee at the first authors' institution (ID No. 687).

[2]Inclusion criteria required participants to rate themselves as 'moderately fit' on a customised screening form for physical activity readiness, but not be undertaking structured resistance exercise, or have specific aesthetic or fitness goals (e.g., fat loss, build muscle, compete in any physical contest, event or sport, etc.). Furthermore, all participants were asymptomatic of low back pain and confirmed that they had never undertaken isolated lumbar extension exercise.

Twenty-six recreationally active[2] asymptomatic males ($n = 14$) and females ($n = 12$) with no previous training experience of the isolated lumbar extensors were recruited (see Table 1 for characteristics). Participants were randomised in a sex counterbalanced fashion to one of two intervention groups; a high load group (HL, $n = 13$), or a low load group (LL, $n = 13$). Prior to testing, participants completed an informed consent and a customized screening form for physical activity readiness. For the purpose of this study, the exclusion criteria included: individuals who were currently undertaking structured resistance exercise, suffered from a heart condition, had a history of chronic low back pain, any contraindications to exercise identified on the customized screening form for physical activity readiness, or any knee or hip conditions which prevented the use of the machine restraints.

Prior to baseline testing, all participants attended a familiarization session where they were introduced to the biomechanics of the ILEX machine. Participants were seated in the ILEX machine with their thighs perpendicular to the seat and a thigh restraint tightened across them to ensure the legs could not lift from the seat. A femur restraint was placed fixed just above the patella and a footboard tightened in driving the participant's shanks upward and back, and thus also the femur. The torque at the distal end of the femurs also resulted in an opposing torque at their proximal end. The combination of these restraints is to prevent upward movement and rotation, of the pelvis whilst permitting full lumbar mobility. The tightness of the restraints was checked by ensuring the participant could not lift their heels form the footboard and ensuring the participant could not rotate their pelvis when leaning forwards and backwards (confirmed by an absence of opposite rotation of a rolling pad at the upper posterior pelvis—see pelvic restraint; Fig. 1). The head rest was
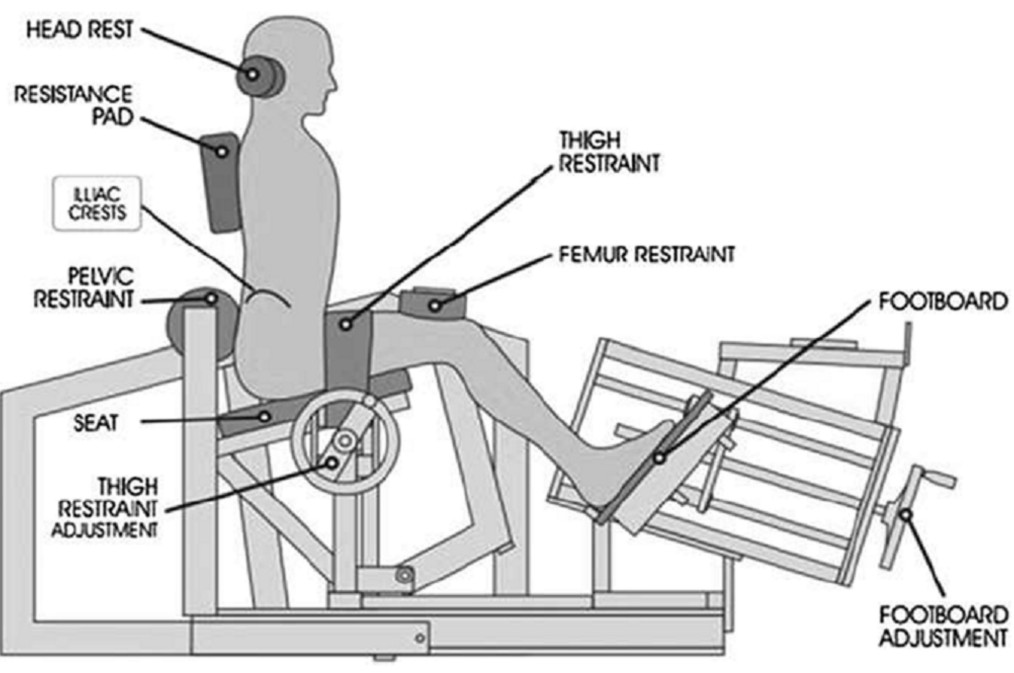

**Figure 1 Restraint system for the MedX Isolated lumbar extension machine.**

[3]RPE-E; 0 = no exertion, 1 = extremely easy, 3 = easy, 5 = somewhat hard, 7 = hard, 9 = very hard, 10 = maximal effort.

[4]RPE-D; 0 = no discomfort, 1 = minor discomfort, 3 = mild discomfort, 5 = moderate discomfort, 7 = severe discomfort, 9 = very severe discomfort, 10 = maximal discomfort.

adjusted accordingly whilst participants were informed that all extension force was to be applied to the resistance pad (see Fig. 1). Participants were assessed to be comfortable and to not be experiencing any pain, numbness, or paraesthesia. Participants were then assessed for lumbar range of motion using a goniometer built in to the ILEX machine (MedX, Ocala, FL, USA), and performed isometric testing using the ILEX machine every 12° beginning at 72° (full lumbar flexion) through 60°, 48°, 36°, 24°, 12°, and 0° (full lumbar extension) to allow them to experience the technique required and reduce any learning effect. Participants were also familiarised with the rating of perceived effort (RPE-E)[3] and rating of perceived discomfort (RPE-D)[4] 0–10 scales. Participants read the description of anchoring for both scales (detailed elsewhere *Fisher, Steele & Smith, 2017*; *Fisher, Ironside & Steele, 2017*; *Steele et al., 2017d*) and confirmed their understanding of differentiation between effort and discomfort.

The testing protocol for the MedX ILEX machine followed the process described above for the familiarization session; participants were seated in the ILEX machine and a thigh and femur restraint tightened to permit full lumbar mobility whilst ensuring the pelvis could not rotate. Participants were provided a specific dynamic warm-up of the lumbar extensors for ~60 s using ~27 kg for males and ~20 kg for females, followed by 3 practice isometric actions at full flexion, mid-range of motion, and full extension using 50% effort. An MVC was then performed at 7 different joint angles (as described above) by the participant gradually building force up to maximal effort over 3 s, and then relax their contraction over a further 3 s. Participants were provided with verbal encouragement to ensure maximal effort and were permitted ~10 s complete recovery between testing angles.

The ILEX machine and restraint system (Fig. 1) prevents rotation of the pelvis allowing training and testing of the lumbar extensors in isolation. The device has high test-retest reliability values of $r = 0.81$–$0.97$ in asymptomatic persons (*Graves et al., 1990*).

Training was conducted at a frequency of 1 day/week for a period of 6 weeks, as has previously been evidenced to significantly improve ILEX strength (*Steele et al., 2015*). Both groups performed a dynamic warm-up of the lumbar extensors for ∼60 s using ∼27 kg for males and ∼20 kg for females. This was followed by a single maximal isometric test at 72° (full lumbar flexion) from which the load equating to 80%- (HL) or 50%- (LL) maximum torque was used for a single set of ILEX exercise. The use of this single MVC allowed weekly prescription and progression of exercise training load throughout the intervention. Repetitions were performed taking at least 2 s to complete the concentric phase, holding for 1 s in full extension and taking at least 4 s for the eccentric phase to ensure standardisation, with visual time feedback on a display in front of the participant. The ILEX machine also provides an audible sound at the completion of each phase of the repetition (e.g., at full flexion—72°, and full extension—0°) to ensure the full range of motion is performed for each repetition. As fatigue incurred, the repetition duration generally increased but participants were encouraged never to move faster than the predetermined repetition duration. The exercise was ceased when a set endpoint of momentary failure had been achieved as per definition from *Steele et al. (2017b)* i.e., despite their maximum effort, participants could not complete the concentric phase of a repetition. A single set of ILEX exercise was chosen due to the similar strength increases to a multiple-set protocol (*Steele et al., 2015*). Immediately following each training session, each participant was asked to report a value for effort (RPE-E) and discomfort (RPE-D) using the aforementioned 0–10 scales that permitted appropriate differentiation of the 2 perceptions (*Fisher, Steele & Smith, 2017*; *Fisher, Ironside & Steele, 2017*; *Steele et al., 2017d*). All groups were asked to refrain from using any other lumbar conditioning exercises for the duration of the study.

The independent variable for analysis was the group (HL or LL) and the dependent variables were change in strength (considered as both peak MVC and a strength index [SI] calculated as the area under the torque curve across the participants range of motion using the trapezoidal method), and RPE-D, time under load, and number of repetitions performed, averaged across the six training sessions. Analysis was not performed for RPE-E since all participants reported maximal effort (i.e., 10) at the cessation of dynamic exercise in both HL and LL. Shapiro–Wilk test was conducted to examine whether data met assumptions of normality of distribution and Levene's test was used to examine assumptions of homogeneity of variance. Analysis of covariance (ANCOVA) was used for between group comparisons of strength changes (post minus pre values) with baseline measures as covariates. Point estimates for change in SI are provided adjusted for baseline scores as covariate, along with the precision of those estimates using 95% confidence intervals (CI) calculated for within group adjusted changes. The 95% CIs can be interpreted further to indicate that significant within group changes occurred if the upper or lower limits do not cross zero. Further, Gardner-Altman plots were produced using Estimation Statistics (*Ho & Claridge-Chang, 2017*) for data visualization. Independent samples *t*- tests were used to compare between conditions for time under load and number of repetitions.

**Table 2  Pre-intervention results, post-intervention results, and changes in strength.**

| Variable | Pre-intervention (Mean ± SD) [95% CIs] | Post-intervention (Mean ± SD) [95% CIs] | Changes (Estimated marginal means [95% CIs]) |
|---|---|---|---|
| SI (Nm degrees) | | | |
| HL | 19,407 ± 5,395 [16,475–22,339] | 22,300 ± 5,665 [19,221–25,379] | 2,891 [1,612–4,169] |
| LL | 18,930 ± 4,245 [16,623–21,237] | 21,790 ± 5,165 [18,981–24,599] | 2,865 [1,587–4,144] |
| MVC (Nm) | | | |
| HL | 344.3 ± 110.16 [284.4–404.2] | 395.1 ± 108.6 [336.1–454.1] | 51.7 [24.4–79.1] |
| LL | 326.2 ± 66.26 [290.2–362.2] | 373.2 ± 74.77 [332.5–413.9] | 46.0 [18.6–73.3] |

**Notes.**

SI, strength index; HL, Heavier load (80% MVC); LL, lighter load (50% MVC); MVC, maximal voluntary contraction; SE, standard error; CIs, confidence intervals; Nm, Newton metres.

RPE-D did not meet assumptions of normality of distribution and so was compared using a Mann Whitney-$U$ test and descriptive statistics are presented as medians and interquartile ranges. All statistical analyses were performed using IBM SPSS Statistics for Windows (version 23; IBM Corp., Portsmouth, Hampshire, UK) and $p < 0.05$ set as the limit for statistical significance.

# RESULTS

Between group comparisons using ANCOVA revealed no significant difference for change in MVC ($F_{(1,23)} = 0.095$, $p = 0.761$) or SI ($F_{(1,23)} = 8.514 * 10^{-4}$, $p = 0.977$). Table 2 shows pre-intervention results, post-intervention results, and estimated marginal means for changes in each outcome measure with 95% CIs for the changes. All changes were considered to be significant within groups as 95% CIs did not cross zero. Figure 2 presents the Gardner-Altman plot with individual responses for change in SI for each group, in addition to the between group difference in change scores with 95% CIs, and Fig. 3 shows the same for change in MVC.

Between group comparisons using an independent $t$-test revealed significantly greater time under load and number of repetitions for LL compared with HL (both $t_{(24)} = -7.865$, $p < 0.001$). Between group comparisons using a Mann Whitney-$U$ test for RPE-D revealed significantly greater values for LL compared with HL ($U = 23.500$, $p = 0.002$). Time under load, number of repetitions, and RPE-D are shown in Table 3.

# DISCUSSION

The present study compared low-volume resistance training of the lumbar extensors using heavier- (HL; 80% MVC) or lighter- (LL; 50% MVC) load ILEX in asymptomatic, recreationally active males and females. The findings from this study serve to support the efficacy of both HL and LL ILEX exercise towards strengthening of the lumbar extensors using a low volume (single-set), low frequency (1×/week) approach.

The present data suggests that there are significant increases in ILEX strength (for both MVC and SI) for HL and LL resistance training, with no between group differences. This is supportive of previous research which suggests that, where tested by impartial means, strength increases are similar between heavier- and lighter-load resistance training when

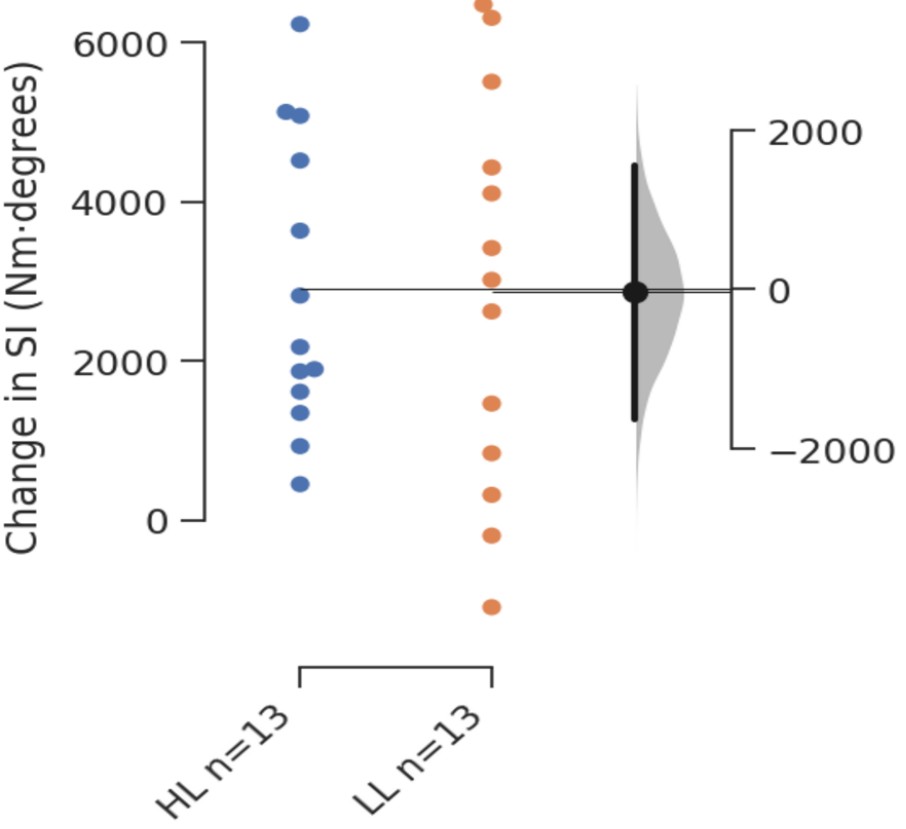

**Figure 2** Gardner-Altman plot of change in SI.

taken to the point of momentary failure (*Fisher, Steele & Smith, 2017*; *Fisher, Ironside & Steele, 2017*; *Schoenfeld et al., 2017*). This similarity of strength increases is hypothesized to result from the maximal synchronous (in the case of HL), or sequential (in the case of LL when exercise is continued to MF), motor unit recruitment (*Fisher, Steele & Smith, 2017*), as per the size principle (*Denny-Brown & Pennybacker, 1938*; *Potvin & Fuglevand, 2017*). The impartial methods used herein are important for strength training practitioners to consider since they serve to minimise skill acquisition and the resultant strength improvements through the practicing of lifting heavy loads. In context, people are likely to seek improvement in ILEX strength for the purpose of reducing the likelihood or severity of chronic low-back pain, and as such we believe that the methods used herein (e.g., dynamic training and isometric testing) are essential for avoiding the confounding effects of skill acquisition in determining whether general strength gains have occurred. Indeed, previous work has actually shown that ILEX strength is associated with lifting capacity (*Reyna Jr et al., 1995*; *Matheson et al., 2002*) which improves after ILEX resistance training (*Mooney et al., 1993*), as does the amount of resistance moved in the deadlift exercise (*Fisher, Bruce-Low & Smith, 2013*). Other studies also suggest that ILEX strength may be related to balance and motor control during gait (*Steele et al., 2014*; *Behennah et al., 2018*). Furthermore, isometric strength increases following dynamic ILEX training are shown to result in a

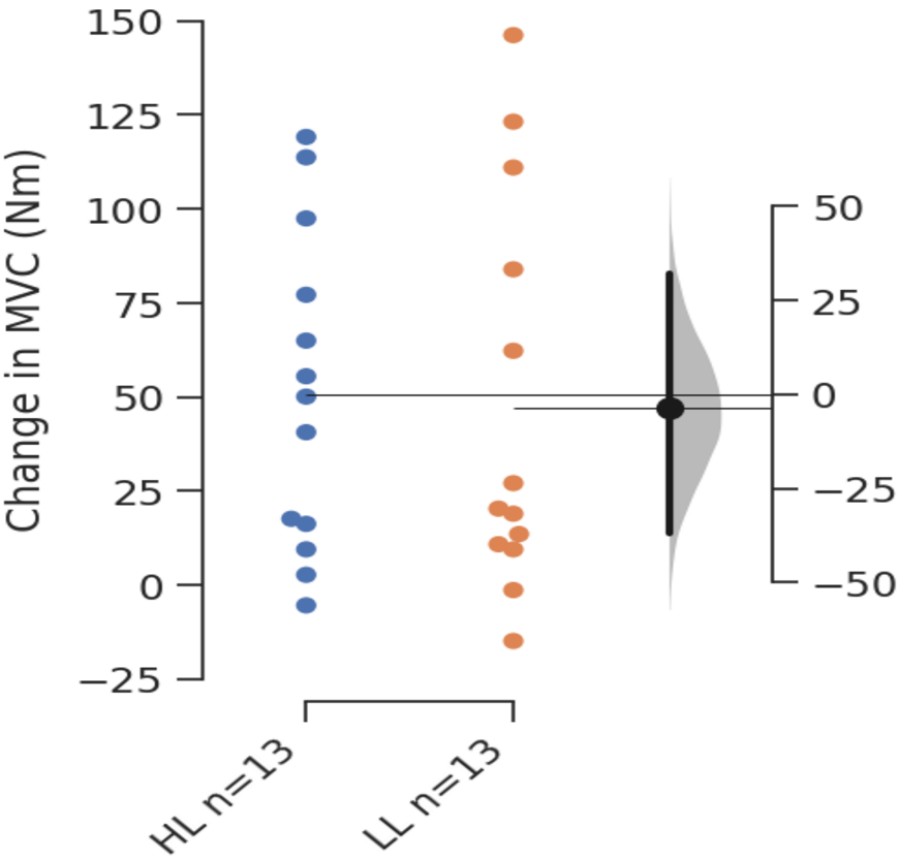

**Figure 3  Gardner-Altman plot of change in MVC.**

**Table 3  Time under load, number of repetitions performed, and RPE-D.**

| Variable | HL ($n = 13$) | LL ($n = 13$) |
|---|---|---|
| Time under load (s) | $50.5 \pm 15.4$ | $158.5 \pm 47.0$ |
| Number of repetitions ($n$) | $8 \pm 3$ | $26 \pm 8$ |
| RPE-D (0–10)* | $4.8 \pm 2.5$ | $7.8 \pm 1.8$ |

**Notes.**
RPE-D is presented as median and interquartile range.
HL, heavier load (80% MVC); LL, lighter load (50% MVC); MVC, maximal voluntary contraction; RPE-D, rating of perceived discomfort.

reduction in chronic low-back pain at both heavier-loads (80% MVC (*Bruce-Low et al.,
2012*; *Steele et al., 2013*; *Steele et al., 2017a*)), and lighter-loads (35–50% MVC (*Helmhout
et al., 2004*; *Harts et al., 2008*; *Risch et al., 1993*)).

As per previous research comparing HL and LL resistance training to momentary
failure (*Fisher, Ironside & Steele, 2017*), participants in the LL group performed a
significantly greater number of repetitions (LL $= 26 \pm 8$ vs. HL $= 8 \pm 3$ repetitions;
$p < 0.001$) resulting in a significantly longer time-under-load (LL $= 158.5 \pm 47.0$ vs. HL $=
50.5 \pm 15.4$ s; $p < 0.001$) compared to the HL group. Maximal effort was confirmed by the

use of an RPE-E scale where all participants in both HL and LL conditions reported maximal values (e.g., 10). Previous research has reported counterintuitive (e.g., submaximal) values for effort measured by traditional RPE scales when participants have performed different exercises at different relative loads to the point of momentary failure (*Shimano et al., 2006*). As discussed, it seems likely that previously most trainees anchored their perceptions of effort upon their perceptions of discomfort (*Steele et al., 2017d*). However, the present study adds to the body of literature suggesting that when instructed, participants can differentiate their perceptions of effort and discomfort (*Fisher, Ironside & Steele, 2017*; *Fisher, Farrow & Steele, 2017*; *Stuart et al., 2018*). The present data supports this previous research; that longer time-under-load results in a greater degree of discomfort, as reported by the participants in the LL compared to the HL condition (LL = 7.8 vs. HL = 4.8, $p$ = 0.002).

Within the present study we did not measure metabolite accumulation or blood-based markers for metabolic stress, however, we have previously hypothesised that the greater values for discomfort resulting from a longer time-under-load are likely a product of a decrease in pH, elevated blood lactate (BLa), cortisol, and inorganic phosphate ($P_i$) along with increases in $H^+$ as a result of the prolonged elevated adenosine triphosphate (ATP) production (*Genner & Weston, 2014*; *Schott, McCully & Rutherford, 1995*; *Takada et al., 2012*; *MacDougall et al., 1999*). *Behm et al. (2002)* suggested that following LL resistance exercise (20RM) fatigue occurred as a result of peripheral factors, potentially a decrease in the contractile strength of muscle fibres and an inability to transmit the impulse across the neuromuscular junction (*Boyas & Guevel, 2011*; *Gandevia, 2001*). Perceived effort is likely centrally mediated thus explaining similar results for HL and LL conditions in the present study, whereas perceptions of discomfort may be more closely associated with afferent feedback (*Marcora, 2009*).

Of course, the present study is not without its limitations. Notably, whilst we have considered strength increases and perceptual responses to HL and LL ILEX exercise, these results in an asymptomatic participant sample might not be applicable to those with non-specific mechanical chronic low-back pain. Symptomatic persons might be predisposed to a greater degree of discomfort through rehabilitation as a result of their low-back pain for both HL and LL exercise. As such future research might consider a similar research design conducted with symptomatic chronic low-back pain patients and compare the perceptual responses between asymptomatic and symptomatic persons, as well as strength and pain reduction improvements. In addition, whilst the use of a regular MVC was appropriate to ensure progressive and specific loading strategies, it might have presented a confounding issue. Since both HL and LL training groups performed a single MVC at 72° (full flexion) each week, this practice of the test by both groups might have converged strength increases between groups. However, this would likely impact the MVC to the greatest degree with potentially minimal impact on the SI data (which was a product of all tested angles). Further research might also consider the longevity of the exercise intervention since previous research has suggested that repeated exposure to exercise conditions known to cause discomfort increases pain tolerance (*O'Leary et al., 2017*).

Finally, we should discuss the sample size of participants. We did not conduct an *a-priori* power analysis but rather collected data throughout the duration of availability of a research assistant. However, post-hoc we have calculated the Cohen's $f$ effect size between two intervention groups in a prior study from our lab (*Steele et al., 2015*) as 0.16—a small effect. Assuming that a moderate effect might be considered meaningful here, G*Power suggests with an ANCOVA model as used herein, that $n = 128$ would be required for power of 0.8 at an alpha of 0.05 to detect an effect of 0.25. As many might perhaps agree, this is an unrealistic number for a study in this field to recruit. As such we should consider that the present study might be underpowered, although from the very similar point estimates and precision of those estimates it seems that any effect, if one were to truly exist, would be very small and practically difficult to justify in terms of meaningfulness.

## CONCLUSION

The present study is the first to compare HL (80% MVC) and LL (50% MVC) resistance training for the lumbar extensors in recreationally active males and females. Our data supports previous research that HL and LL resistance training produce similar chronic strength adaptations. With the above in mind, it is important to consider that a historical approach to training the muscles of the trunk has been to use lighter loads and higher repetition ranges. This is likely because of the predominance of type I muscle fibres (*Thorstensson & Carlson, 1987*), and the resulting repeated lower force actions needed for postural control and stability. However, based on the evidence presented herein as well as previous literature (*Fisher, Ironside & Steele, 2017*; *Fisher, Farrow & Steele, 2017*; *Stuart et al., 2018*) this lighter-load exercise likely produces a greater degree of discomfort which might result in persons failing to perform higher effort exercise (e.g., reaching MF) in real world settings, and could result in suboptimal adaptations over a training intervention. This is particularly important since poor lumbar strength is associated with low back pain. The present data also adds to the body of literature that low-volume (single-set), and low frequency (1 day/week) ILEX exercise performed to momentary failure produces significant strength increases. As such, we suggest that this minimal dose approach to ILEX exercise might be manageable for persons to increase their ILEX strength with the aim of reducing risk of low back injury and pain (*Steele, Bruce-Low & Smith, 2015b*).

### Funding
The authors received no funding for this work.

### Competing Interests
The authors declare there are no competing interests.

## Author Contributions

- James P. Fisher conceived and designed the experiments, performed the experiments, contributed reagents/materials/analysis tools, prepared figures and/or tables, authored or reviewed drafts of the paper, approved the final draft.
- Charlotte Stuart conceived and designed the experiments, performed the experiments, authored or reviewed drafts of the paper, approved the final draft.
- James Steele conceived and designed the experiments, performed the experiments, analyzed the data, contributed reagents/materials/analysis tools, prepared figures and/or tables, authored or reviewed drafts of the paper, approved the final draft.
- Paulo Gentil analyzed the data, contributed reagents/materials/analysis tools, prepared figures and/or tables, authored or reviewed drafts of the paper, approved the final draft.
- Jürgen Giessing contributed reagents/materials/analysis tools, prepared figures and/or tables, authored or reviewed drafts of the paper, approved the final draft.

## Human Ethics

The following information was supplied relating to ethical approvals (i.e., approving body and any reference numbers):

Southampton Solent University, Health Exercise and Sport Sciences ethical review board granted ethical approval for this research (#687).

## Data Availability

The raw data are provided in the Supplemental Files.

## Supplemental Information

Supplemental information for this article can be found online at http://dx.doi.org/10.7717/peerj.6001#supplemental-information.

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
