# Peer review of "Heavier- and lighter-load isolated lumbar extension resistance training produce similar strength increases, but different perceptual responses, in healthy males and females"

_PeerJ, doi:10.7717/peerj.6001_

## Round 0.1 · original submission · Major Revisions

Dear authors:

Your manuscript was evaluated by expert reviewers. Please address reviewer´s concerns in detail.

However, I must warn you that your work was strongly criticized by reviewers, and I have serious doubts regarding this work and its meaningfulness. With this in mind, please, you and your team must achieve a remarkable sophistication of the work and succeed in answering the concerns of the reviewers.

Thank you very much.

Sincerely,
Rodrigo Ramírez-Campillo
Academic Editor
PeerJ

·

Basic reporting

NA

Experimental design

NA

Validity of the findings

NA

Additional comments

GENERAL COMMENTS

Overall, this is an interesting study that adds value to the literature. However, there a few serious problems that needs to be improved before the manuscript is approved for publication.

First, the manuscript is written with a clinical orientation despite the fact that participants were healthy. This is misleading and should be changed throughout.

Second, there is a serious confounder in this study. Participants practiced the test on each and every session with the MVCs. This needs to be acknowledged and discussed as a limitation and as a possible reason accounting for the lack of differences between the groups.

Third, some of the results are presented in an unclear manner. Specifically, Table 2 can be improved and graphs are needed to illustrate the individual responses in both groups for the main outcome measures.

Forth, there are a number of overreaching conclusions and mismatches between the claims and the citations.

SPECIFIC COMMENTS

ABSTRACT

Line 40 – Add “to momentary failure.” After MVC.

Lines 40-43 – Add an indication of effect size. Maybe CIs of differences for the main outcomes or even percent difference. P values are not enough.

Line 43 – I would say that the present study “shows” rather than “supports”.

Lines 44-47 – You need to change the practical suggestion part as is not supported by the data. You tested healthy, uninjured participants. It remains unclear if the results of the present study will persist with CLBP patients.

INTRODUCTION

The message of the first paragraph is not fully agreed upon. I suggest writing it in a more balanced and suggestive tone and possibly discuss the conflicting perspectives. Also, I believe that citing Schoenfeld’s meta-analysis on reaching failure is important and relevant.

The second paragraph begins with a detailed discussion on perception of effort and discomfort but it is unclear what these perceptions are as they have not been defined. Begin this paragraph with a brief introduction to these scales and their definitions. The rest of the paragraph would then be easier to understand and follow.

I wonder if you can reduce the amount of abbreviations. Already, by the second paragraph, excluding the physiological abbreviations (e.g., lactate), I counted four. Personally, I find it harder to read a paper this way.

Lines 75-80 - I disagree with the way you set up setting up the research question around rehabilitation. You did not test injured participants and thus the results do not generalize to such populations.

Lines 81-96 – This paragraph is somewhat confusing. The topics seem to change suddenly without flow. Perhaps just discuss the comparison studies and their limitations and remove the sentences dealing with causes of CLBP.

Line 90 – Morton did not directly confirm that all fibers are activated when going to failure.

Line 92 – I am not sure what the purpose of citing it and discussing the study by Risch et al as based on this sentence, no comparisons to other loads were made and low loads were used. I fail to see the relevance here.

Lines 93-96 - Based on the short literature review in this paragraph, I am left confused as to what it is showing us and to what holes in the literature it points to. It seems as if studies that used low loads and did not implement failure led to beneficial effects.

Lines 97-101 – I would definitely use softer language. “Often almost completely passive” is too strong and likely inaccurate. Physiotherapists regularly use exercises and there are indeed many studies showing effectiveness of exercise to CLBP with ILEX RT being one of them. I would set up the questions in a “fairer” way: exercise seems to be effective to reduce CLBP, with ILEX RT being one strategy found to be useful.

Line 102 – I would make a clearer distinction between prevention (which I think is a more suitable angle for your study given that you examined healthy adults) and pain reduction to those already suffering from CLBP. While the distinction is there, I think it needs to be emphasized more. But again, I disagree with the strong clinical emphasis.

Line 109 – “reaching true MF and incurring the desired adaptations over a training intervention.” Since it is not mandatory to reach MF to achieve desired adaptions, I would soften this sentence and make it more suggestive.

METHODS

Line 122 – Did you conduct a power analysis? Please provide justifications/explanations for the utilized sample size.

Can you please provide more details about the subjects training background? Given that this is a RT study, more information about their RT experience is required.

Line 136 – Given that the descriptions are likely composed of a few sentences, include them in the manuscript rather than referring to reader to other articles.

Lines 142-145- It says that “…building force up to maximal effort over 3 seconds, and then relaxing over a further 3 seconds.” And then that “…and were permitted ~10 seconds recovery between testing angles.” Is it 3 or 10 s of rests between MVCs? This needs to be clarified.

Lines 149-170 –Did the MVC tests at every angle take place on the same day and prior to the single set to failure? Were they conducted every session? How did you select the exact % of MVC given that the presumably the forces differed at each angle? Which angle (or what is the average?) served as the anchor?

Line 151 – “…due to the potential for overtraining when the lumbar extensor muscles are isolated.” The citation provided to justify this rationale does not concern overtraining. It was a reliability study. I find it very difficult to believe that there is a risk of overtraining if participates complete more than a single training session involving a single set a week. Unless direct evidence is available to support this claim remove this sentence.

RESULTS

Table 2 is unclear. First, is the variability presented as SE or SD? This needs to be clarified in all columns. In case the data is presented as SE in the pre-interventions values please change it to SD. Second, please include a post-intervention column to the right of the pre-intervention column with means and SD. Third, it is unclear why the 95%CIs of the differences aren’t enough here? Including both the estimate marginal means (±SE) for changes in each outcome measure, and 95%CIs for the changes is confusing and redundant. I would include the 95%CIs of the within subjects under the means and SD of the pre and post intervention columns and include a separate column for the change score 95%CIs to make the results easier to interpret.

Please prepare a graph of some sort to visually present the individual responses (pre-post changes) for MVCs and SI. Estimation plots would be ideal.

DISCUSSION

Line 202 – I am not sure why this study is introduced and discussed solely in view of rehabilitation whereas the population was healthy. I find this misleading. Anything to do with rehabilitation is speculation and at most should be discussed as a secondary possibility after discussing the results for what they are: training with low and high loads with the ILEX seem to lead to similar strength adaptions among recreationally active subjects over a six week period using only a single set to failure once a week. Thus, professionals can let their trainees and clients choose the loads as the strength adaptions seem to be similar when using this particular protocol.

Lines 209-219 – This is not accurate as subjects trained the task once a week during the MVCs tests. Thus, and similar to the results reported by Morton, the lack of differences observed between groups may stem from the fact that subjects trained the test with each and every training session. I think that you introduced, rather than controlled for the very confounder you aimed to avoid with the regular MVC testing. This is a considerable confounder of this study and should be discussed as such in the limitation section.

Lines 212-2016 – I am not sure that this sentence adds to your discussion and suggest removing it.

Lines 244-248 – Most studies on central fatigue actually reported that it plays a larger role in submaximal extended efforts. I wonder if this is too speculative to discuss central fatigue here. Another possible pathway is psychological. It could be that with lighter loads the end point of a set is considerably further away than heavier loads which could influence the perception of difficulty.
In the limitations section I strong suggest you mention the confounder discussed above: the fact the MVCs were tested before each testing sessions allowing subjects to gain experience in how to conduct the test which could have confounded/reduced the real differences between conditions. Furthermore, while you did cite articles showing relationships between ILEX and other, more common exercises that practitioners have access to, I think it is fair to mention the reduced degree of external validity as very few practitioners have access to ILEX and thus the degree of generalizability is reduced.

CONCLUSION

First, as mentioned before, the study was not done on those with low back pain so the discussing should not revolve around clinical populations. Second, while reaching failure is discussed throughout the manuscript as a key and very important variable, studies show that reaching failure is not mandatory to achieve positive and similar strength increases when volume high enough. Thus, to avoid having the readers assume that this is the case, it will be beneficial to touch on this fact in the introduction.

Lines 268-272- Perhaps emphasizes what is unique about this study compared to the others you cited. At present, it is unclear.

·

Basic reporting

This is a well-written manuscript with good use of language. There are a few points of correction needed in regards to grammar and punctuation. Please see the specific comments.

Experimental design

Clear and sound with only a couple of suggestions as indicated below. The major point of clarification needed is around the statistics for the primary measure. Please see the specific comments.

Validity of the findings

See specific comments for a few issues around limiting conclusions to data presented and clarify the issues with statistics.

Additional comments

Specific comments
50-58 - first sentence is out of place. I would suggest leading with the second sentence and moving the content of the first sentence to later in the paragraph. It doesn’t seem appropriate to lead with motor unit recruitment. The remainder of the paragraph doesn’t flow well. Can this paragraph be re-written to better explain the main point?
60-64 – please explain the RPE anchoring in more detail and define it at this point.
82-84 – while this may be true, there is also a contrary body of literature around pain and strength. I do agree with the rehab implications of your work, but perhaps it could be framed around optimizing rehabilitation approaches, rather than on the relationship between strength/endurance and pain? In addition, I would highlight your group’s previous review (Sports Med, 2016) stating that the lumbar extensors appear to be one of the only muscle groups that would benefit from direct isolation training over and above that that occurs with compound training (at least for strength measures).
113-115 – Poorly worded – If all of the outcome measures can be considered “primary”, they should be stated together with strength. If not, please reword to include primary and secondary outcomes.
126 – reference for the PARQ
136 – as suggested for the introduction, it would benefit readers to have a simple description of the anchoring process here and appropriate references to the scales themselves
178-179 – I would ask the authors to explain their rationale for using the change scores as the DV in the ANCOVA. In general, change scores are quite liberal (even when controlling for baseline measures as you have) compared with using the actual raw values in an ANCOVA. Were the baseline measures between groups significantly different? If so, using the change scores assumes a similar degree of adaptation would occur between groups, which may not be true. Lower strength/endurance levels would likely adapt to a greater degree to a similar stimulus. The other side of this is that the higher baseline scores (if changing similarly) would result in a lower % change than the group with the lower baseline scores. All of these factors lend me to question the validity of this approach for a training study. In my opinion, the more robust method would be to use the actual scores in the ANCOVA (still controlling for baseline). Please consider these points, provide your rationale, and/or adjust your statistics accordingly.
All other statistics appear to be appropriate and presented well.
206 – I’m not sure of the relevance of the phrase “when tested by impartial means”. Can you please clarify the meaning and importance of this?
208 – Please revise “This is hypothesized” to “The similarity of strength increases is hypothesized”
208 – 209 – This is a vague sentence. It is unclear what is being referred to with the addition of the word “respectively” as to HL and LL.
209 – 211 – The reference to “impartial methods” again here is unclear. You’ve emphasized this a few times, but your methods weren’t designed to compare such methods with others. Please temper the language and conclusions around this throughout the manuscript.
212 – 216 – Practically, I don’t disagree with these statements. Having said that, these statements neither seek to explain or interpret your findings of this study, nor are referenced appropriately. Please refocus and revise.
244 – change is to it
244-247 – Please add punctuation to the following sentence as follows: “As such, it appears that when performing HL resistance exercise, cessation occurs as a result of central fatigue; whereas, during LL resistance exercise, peripheral fatigue is responsible for an inability to transmit the impulse across the neuromuscular junction.”
244-247 – Please provide references for each aspect of this sentence. Check to ensure the accuracy of these statements as well, as I don’t believe the literature shows that this is definitive.
260 – comma needed between “mind” and “it”. There are several instances throughout the manuscript where similar such commas are missing.
References – please check for consistency. There are several citations referenced with periods at the end, and several without, for example.
Tables – please provide footnotes for acronyms and abbreviations for each table

---

## Round 0.2 · Minor Revisions

Dear authors,

Although the reviewer 1 recommended acceptance of your manuscript, some important observations were noted. Therefore, authors should take these into consideration.

L.47 - consider adding some raw values or CIs in addition to the p value.

L. 263 - The study by Behm does not suggest this as no differences in muscle activation were observed between HL and LL. However, the 20RM did lead to greater peripheral fatigue. Consider modifying the part about central activation.

Table 2 - I think you need to change the second column from pre- to post-intervention.

Concerning the the sample size issue, I see where you are coming from and mostly agree but think you still need to mention something about it in the text. The fact that you didn't conduct a power-analysis is not a reason not to discuss, justify and defend your decisions in the text as you did in your response to me. In my opinion is it important to include the points to brought up in the text and acknowledge the possibility of the study being under-powered.

·

Basic reporting

NA

Experimental design

NA

Validity of the findings

NA

Additional comments

Excellent job addressing my comments. The paper reads much better now. Congratulations.

A few additional points:

L.47 - consider adding some raw values or CIs in addition to the p value.

L. 263 - The study by Behm does not suggest this as no differences in muscle activation were observed between HL and LL. However, the 20RM did lead to greater peripheral fatigue. Consider modifying the part about central activation.

Table 2 - I think you need to change the second column from pre- to post-intervention.

Concerning the the sample size issue, I see where you are coming from and mostly agree but think you still need to mention something about it in the text. The fact that you didn't conduct a power-analysis is not a reason not to discuss, justify and defend your decisions in the text as you did in your response to me. In my opinion is it important to include the points to brought up in the text and acknowledge the possibility of the study being under-powered.

·

Basic reporting

See below

Experimental design

See below

Validity of the findings

See below

Additional comments

The authors have revised this manuscript in great detail, and have sufficiently addressed the majority of my concerns. There are only a few further points of clarification:
1. I appreciate the detail now used around the anchoring of your RPE scales. This reads very well. My only comment is around the use of the superscript numerals to denote reference to the footnotes. These superscripts are not distinguishable from the reference citations. Can they be presented in another way?
2. Footnote 2 reads "Inclusion criteria required participants to self-rate themselves...." This should be either "rate themselves" or "self-rate".
3. My previous comment about the PARQ was only that if it was the PARQ published by CSEP, it should be referenced as such. If it is not, perhaps indicate it is a "customized screening form for physical activity readiness" or something like that to avoid confusion with the published version.
4. Column 3 in Table 2 is labelled pre-intervention, but I believe this should be post-intervention. Please check.

Once again, I would like to commend the authors on a highly relevant study, which has now substantially improved.

---

## Round 0.3 · accepted · Accept

Congratulations on a well conducted research!

·

Basic reporting

N/A

Experimental design

N/A

Validity of the findings

N/A

Additional comments

This manuscript is acceptable for publication.